# Treatment of Ebola Virus Disease: From Serotherapy to the Use of Monoclonal Antibodies

**DOI:** 10.3390/antib14010022

**Published:** 2025-03-05

**Authors:** Dmitriy N. Shcherbakov, Anastasiya A. Isaeva, Egor A. Mustaev

**Affiliations:** 1State Research Center of Virology and Biotechnology VECTOR, Rospotrebnadzor, Koltsovo 630559, Russia; anastasya_isaeva_1993@mail.ru; 2Department of Natural Sciences, Novosibirsk State University, Pirogova st., 2, Novosibirsk 630090, Russia; e.mustaev@g.nsu.ru; 3Synchrotron Radiation Facility—Siberian Circular Photon Source “SKlF” Boreskov Institute of Catalysis of Siberian Branch of the Russian Academy of Sciences, Nikolskiy pr-t, 1, Koltsovo 630559, Russia

**Keywords:** Ebola virus disease, filoviruses, passive immunization, neutralizing antibodies, monoclonal antibodies

## Abstract

Ebola virus disease (EVD) is an acute illness with a high-case fatality rate (CFR) caused by an RNA virus belonging to the Filoviridae family. Over the past 50 years, regular EVD outbreaks have been reported. The West African EVD outbreak of 2013–2016 proved to be significantly more widespread and complex than previous ones, resulting in approximately 11,000 deaths. A coordinated international effort was required to bring the outbreak under control. One of the main challenges faced by clinicians and researchers combating EVD was the absence of vaccines and preventive treatments. Only recently have efforts led to the development of effective therapeutic options. Among these, monoclonal antibody-based drugs have emerged as the most promising agents for the urgent treatment of EVD. This article aims to review the key milestones in the development of antibody-based therapies for EVD, tracing the journey from the use of convalescent serum to the creation of effective monoclonal antibody-based drugs and their combinations.

## 1. Introduction

For the past four years, the efforts of the scientific community and the healthcare system have been aimed at treating the SARS-CoV-2 pandemic. However, regular Ebola virus disease (EVD) outbreaks [1], the most recent of which was reported in April 2022, require equal attention (Figure 1). Ebolaviruses are single-stranded negative-sense RNA viruses. The virion is formed by seven proteins: nucleoprotein NP, polymerase cofactor VP35, matrix protein VP40, glycoprotein GP, transcription factor VP30, nucleocapsid-associated protein VP24, and RNA-directed RNA polymerase L [2]. NP, VP30, L, and VP35, together with RNA, form a ribonucleoprotein complex necessary for the transcription and replication of the genome [3]. VP40 is able to penetrate lipid bilayers and initiate membrane curvature, leading to the formation of viral particles. VP24 blocks cellular immunity, regulates the synthesis of viral RNA, and is involved in genome packaging [4]. GP plays a key role in virus penetration and pathogenesis.

Ebolaviruses belong to the family *Filoviridae*, which comprises eight virus genus, including *Orthoebolavirus*, *Orthomarburgvirus*, *Cuevavirus*, *Oblavirus*, *Dianlovirus*, *Striavirus*, *Tapjovirus* and *Thamnovirus*. In the genus *Orthoebolavirus* six species that each have a single type of virus have been identified to date: *Orthoebolavirus zairense* with the Ebola virus (EBOV), *Orthoebolavirus sudanense* with the Sudan virus (SUDV), *Orthoebolavirus bundibugyoense* with the Bundibugyo virus (BDBV), *Orthoebolavirus taiense* with the Taï Forest virus (TAFV), *Orthoebolavirus restonense* with the Reston virus (RESTV), and *Orthoebolavirus bombaliense* with the Bombali virus (BOMV) [2].

Despite the existence of The Food and Drug Administration (FDA)-approved vaccines (rVSV-ZEBOV) and drugs (Ebanga, Inmazeb), research should be continued in order to improve both drug effectiveness and availability in regions with the highest disease burden. It is important to note that the above drugs act only against EBOV. However, the BDBV and SUDV also pose a threat to humans, which makes the search for broadly neutralizing antibodies and the development of pan-ebolavirus drugs based on these antibodies an urgent scientific problem [5].

Since neutralizing antibodies are targeted at the GP of ebolaviruses, we will consider its structure in more detail. The image of GP with the main targets for antibodies is shown in Figure 2. GP is a trimer consisting of heterodimers. Each heterodimer contains a GP1 subunit (approximately 470 amino acid residues) and a GP2 subunit (approximately 175 amino acid residues) [6]. The GP1 subunit is responsible for cell attachment and can be divided into three subdomains: the base, head, and glycan cap (GC). GP1 also contains a mucin-like domain (MLD) in the C-terminus, a region with several conserved cysteine residues at the N-terminus, and a receptor binding site (RBS) [7]. The GP2 subunit is responsible for the fusion of the viral and host cell membranes. GP2 contains conserved cysteine residues, an internal fusion loop (IFL), Heptad repeats 1 and 2, an alpha-helical transmembrane domain, and a region homologous to the conserved immunosuppressive motif found in oncogenic retroviruses [8]. Most epitopes are located at the GP1–GP2 interface, GC, and GP “base” (Appendix A).

In this review, we attempted to cover as fully as possible the information concerning serum therapy and specific monoclonal antibodies since the discovery of the virus. The review was performed on all available experimental articles from 1977 to the present. In the case of several articles devoted to one monoclonal antibody, the article that most thoroughly described the structure and properties of the antibody was selected for the review.

## 2. Passive Immunization

Passive immunization is a method of immune defense based on the administration of a set of antibodies developed in response to infection in other organisms. The approach has been successfully used to treat a series of diseases since its discovery by von Behring [9]. The advent of the era of antibiotics and vaccines reduced the value of this method for therapy and prevention. However, it is still used nowadays. This method seems to be useful in combating EVD [10]. It is affordable, while the use of convalescent serum provides a high likelihood of anti-viral protection. This approach was used for urgent EVD therapy.

In 1976, a Porton Down researcher was working with samples from guinea pigs infected with the Marburg-like virus (a recently isolated virus, which was later called Ebola (EBOV); the first name of the virus was due to its morphological and clinical similarity with the Marburg virus (MARV) discovered in 1967) and pricked his finger through the glove. The researcher was placed in quarantine. Interferon was administered 12 h later. However, after the deterioration of the patient’s condition, a decision was made to administer the serum from a convalescent donor from Zaire for the next three days. A total of 450 mL of the serum was injected following testing for a series of pathogens. The procedure was repeated for three days. The researcher recovered after two weeks of illness. One cannot claim that it was the serum transfer that caused recovery since the serum was injected simultaneously with interferon. However, interferon therapy is ineffective in the acute disease phase, while each injection of convalescent serum resulted in a significant (by three orders of magnitude after the first administration) decrease in the viral load [11]. Additional information about the study can be found in Appendix A.

The next documented use of convalescent blood was in 1995 during the Kikwit outbreak in the Democratic Republic of the Congo (DRC). Blood from five convalescent individuals was transfused into eight people diagnosed with EVD; seven out of eight people survived. A comparison of the case fatality rate (12.5%) with the overall fatality rate for the outbreak (80%) suggests the effectiveness of the method [12] (Appendix A). Meanwhile, a detailed analysis of this case does not allow one to make such an unbiased conclusion. In some patients, blood transfusion was not conducted immediately after infection diagnosis but after a certain period, which in some cases equaled two weeks; this cast reasonable doubts on the contribution of the convalescent serum to patient recovery [13].

Controversial results on the use of passive immunization to treat EVD were obtained during the outbreak in 2014–2015. A total of 58 (69%) out of 84 individuals with confirmed EVD survived after the transfusion of two doses of convalescent serum. The survival rate in the control group (no serum transfusion) was 62% (Appendix A). A possible explanation for this could be the absence of the titer of neutralizing antibodies. However, this did not change the observed picture radically [14].

Passive immunization has also been studied in animal models. A hyperimmune equine anti-EVD serum was obtained at the Central Research Institute of the Ministry of Defense of the Russian Federation No. 48 [15]. A series of immunization cycles with live viruses in horses resulted in a virus-neutralizing titer of 1:4096. After purification and fractionation, this preparation was administered to hamadryas baboons two days before infection (group 1) and simultaneously with infection (group 2). All control monkeys (group 3) died, while only one animal receiving antibody preparation together with infection died in the experimental group [16]. Further studies on hyperimmune serum showed that a positive effect required the earliest possible administration (1 h after/before infection) and significantly high titers of neutralizing antibodies [17]. But at the same time, the study of this preparation by Jahrling et al. demonstrated that it is impossible to prevent death in monkeys if the drug is administered on the day of infection [18]. Double administration of the preparation in mice and monkeys also resulted in animal death. However, it should be noted that high viral doses were used (30 median lethal doses (LD50) in mice and 10^4^ plaque-forming units (pfu) were used in monkeys). The protective effect of the drug based on hyperimmune equine serum was observed in guinea pigs only [19]. Another approach to obtaining serum antibody preparation involved immunization with a DNA vaccine followed by the booster administration of flavivirus replicon-based virus-like particles (VLPs). This approach yielded an antibody preparation that provided 100% protection in cynomolgus macaques [20]. Positive results were achieved by Zheng et al. when using an antibody preparation based on equine serum. EBOV VLPs containing Viral Protein 40 kDa (VP40) and glycoprotein (GP) were used for immunization. The administration of the resulting antibody preparation in mice and guinea pigs provided animal protection against 10^3^ LD50 of the virus [21]. Detailed information on the above studies is provided in Appendix A.

The production of antibody preparations based on goat and sheep serum is described. No adverse reactions were observed when it was administered to animals or humans [22]. A similar approach was used by Dowall et al. [23]. The researchers also used ovine serum and EBOV GP but lacked the transmembrane domain for immunization. Nevertheless, the resulting preparation, which was called EBOTAb, had neutralizing activity and could prevent animal death upon administration in guinea pigs both 6 and 24 h after infection. The later administration of the preparation resulted in decreased survival rates. The single injection of EBOTAb on day 3 after infection reduced the survival rate to 33%. Multiple injections of EBOTAb improved the clinical picture: the survival rate increased to 83.3%. However, if the treatment with multiple injections was started on days 4 and 5, the survival rate decreased to 50% and 33.3%, respectively (Appendix A) [23,24]. The study of this preparation in nonhuman primates (NHPs) can be considered successful, even considering the fact that the 100% survival rate was only achieved if the treatment was started on the first day after infection (340 mg per animal daily for 5 days, followed by three subsequent treatment courses every other day). An up to three-day delay in the start of treatment reduced the survival rate to 25% (Appendix A) [25].

Dye et al. proposed an interesting approach that made it possible to obtain a polyclonal antibody-based preparation for passive immunization. Their approach was based on the use of transchromosomal cattle, whose antibody repertoire consists of human immunoglobulins only, to treat EVD. In order to obtain high titers of neutralizing antibodies (1:10,000), cows were immunized with a recombinant GP of EBOV eight times. The single administration of this preparation (100 mg/kg) in mice led to almost 90% survival (Appendix A) [26]. The induction of the generation of neutralizing antibodies by DNA immunization led to lower antibody levels (1:160); however, this antibody preparation also conferred a protective effect in mice [27].

Positive results were obtained upon the transfusion of antibodies generated in response to the subcutaneous immunization of mice with a mouse-adapted EBOV strain. After titration for specific antibodies by the enzyme-linked immunosorbent assay (ELISA) to a titer of 105, serum preparations were used for the passive immunization of naive mice. The authors showed that only high concentrations of antibody preparations provided protection. Preparations with an antibody titer of ≥1:6400 completely protected mice against 10^3^ pfu of the virus. The pivotal role of antibodies in protection against the virus was confirmed by a differential analysis of the protective properties of the immunoglobulin serum fraction. Injections of purified antibodies provided a protection level similar to that of serum administration. The authors noted the importance of using a homologous serum, which, in their opinion, was responsible for the survival of all experimental animals, even in the case when serum administration was started two days after symptom onset. Repeated administration of the virus after 77 days did not result in any EBOV symptoms, while animals developed their own immunity. Moreover, the antibody preparation also provided protection upon passive transfer to severe combined immunodeficient mice (SCID) [28].

At the same time, a similar approach implemented by Jahrling et al. was unsuccessful (Appendix A) [29]. The authors transfused the serum isolated from three EBOV-immune primates. The first primate was vaccinated by mouse-adapted EBOV, which had a weak effect on NHPs. The second primate was treated with a γ-radiated whole-virione vaccine against EBOV. The third animal presented a rare case of survival after virulent EBOV infection. This result may be due to the fact that ELISA titers often inversely correlate with plaque neutralization assay (PRNT) titers [30]. This phenomenon was influenced by the fact that immunization with inactivated antigens from immune animals resulted in low PRNT titers but high ELISA titers and vice versa: the administration of the live virus in immune animals resulted in a pronounced neutralizing effect at low ELISA titers. Passive immunization was conducted immediately after infection. Despite the presence of a significant concentration of specific antibodies in recipient monkeys (up to 1:3200 as determined by ELISA but not PRNT), all animals died.

Negative results were obtained upon passive immunization using serum from monkeys that survived after the study of an anti-viral drug against EBOV and SUDV. Despite the high titers (1:6400–1:100,000) of specific antibodies in the ELISA test and double administration of the drug, all animals, except for one, died. The only primate that survived belonged to the group receiving serum from a survivor of SUDV infection [31]. A similar experiment, which was based on the same principle but differed in crucial details, was conducted by Dye et al. and led to positive results. A high-titer serum from macaques who survived immunization with a candidate DNA vaccine followed by EBOV infection was obtained at the first stage. In the first experiment, the serum was administered almost simultaneously with infection (15–30 min after infection) using 10^3^ pfu of MARV. The serum was injected thrice in order to maintain a high antibody level. All animals survived in the absence of any disease signs. The secondary administration of the virus also did not lead to infection: monkeys developed their own immunity. In the second experiment, which involved both MARV and EBOV viruses, the serum was administered after 48 h. Despite the delay, animals also survived. Only one animal had mild fever symptoms [32]. Studies are described in detail in Appendix A.

Summarizing the results of passive immunization, we would like to point out a number of important aspects. Serum transfer from an organism effectively responding to filovirus infection can protect an intact organism. Failures were usually due to the use of heterologous serum preparations [17,18,19,28] and insufficient concentrations of antibodies that were administered once [11,18,33]. On the contrary, the use of homologous serum with high concentrations of specific antibodies provided a positive result [11,28,32]. The method of antibody production also plays an important role. Immunization with the inactivated virus appears to induce antibody production with adverse effects such as antibody-dependent enhancement (ADE). The best results are achieved with the use of VLPs and DNA for the induction of specific antibodies [20,21]. This pattern can be explained by the specific aspects of antigen presentation to the immune system: the use of VLPs and DNA does not disrupt the structure of viral proteins, as in the case of inactivation.

The role of monoclonal antibody-based preparations increases in the therapy of various diseases due to the creation and introduction of novel methods for the search and production of these antibodies [34]. Viral diseases are not the exception.

## 3. Ebola Virus-Neutralizing Antibodies

The first monoclonal antibodies against EBOV were obtained by Maruyama et al. in 1999 [35,36]. Antibodies were selected from two phage libraries. The first one was obtained based on the brain marrow sample from two humans who survived the Ebola outbreak in DRC in 1995; the second one was generated based on 10 serum samples from individuals who survived the same outbreak. Eight Fab fragments specifically binding to different viral proteins were isolated using affinity selection. Only one antibody, namely KZ52, exerted moderate neutralizing activity. This antibody could bind to GPs on the viral surface. The remaining antibodies were bound to NP and the soluble form of viral GP.

The ability of KZ52 to protect against EBOV was studied in the guinea pig model. The administration of 50 mg/kg (animal weight) of KZ52, which equals approximately 1:200 in the neutralization reaction, made it possible to avoid animal death when using a viral dose of 10^4^ pfu; however, the immunity was not sterile, and the virus was detected in the animal blood [37]. The use of Ebola pseudoviruses demonstrated that this antibody neutralizes 90% of viral particles at a concentration of 1 µg/mL [38] (Appendix A). At the same time, the injection of 50 µg/kg of the antibody in rhesus macaques did not provide any protection against the virus: all animals died [39]. The mechanism used by KZ52 for virus neutralization has been elucidated: the antibody inhibited the key process responsible for the cleavage of a surface protein by cathepsin via its interaction with the region at the GP1–GP2 interface, which is crucial for virus penetration from the endosome to the cell [38,40,41]. More information on the interaction between KZ52 and EBOV GP and antibody effectiveness is provided in Appendix A.

The phage display technology was also used to produce antibodies from the brain marrow of rhesus macaques that survived after immunization with a high viral dose. The resulting library contained 10^6^ individual phage clones. Selection using the inactivated virus yielded a panel of three Fab fragments: JP3-K9, JP3-K11, and JP3-K14 [36]. Shedlock et al. established the fact that the antibody JP3-K11 utilizes a mechanism different from the one used by KZ52 for neutralization. While KZ52 uses endosomal proteolysis for EBOV neutralization, the principle of JP3-K11 action is based on the conventional mechanism of neutralization through the inhibition of GP recruitment to the receptor [38] (Appendix A).

A study reporting 14 murine monoclonal antibodies obtained using the hybridoma technology was published in 2000 [42]. Despite the fact that 10 out of 14 antibodies provided protection upon simultaneous administration with the virus, with some of them exerting the effect even when injected after 48 h, none of the antibodies had virus-neutralizing activity. Only antibodies a6D8 and 13C6 had moderate neutralizing activity in the presence of complement proteins (80%; 6.25 µg/mL). Furthermore, 13C6 binds to soluble GP with higher affinity but not to the surface GP. The use of alanine scanning made it possible to study the epitopes of 13C6 and 6D8 in detail. Antibody 13C6 was shown to interact with the short fragment of GC, while 6D8 binds to the MLD [43] (Appendix A).

The authors [7,44] further studied antibodies 13F6-1-2 and 14G7, which were obtained in [42]. It was shown that 13F6-1-2 consists of a rare Vλx light chain [7]. The 14G7 paratope structure, on the contrary, does not have any specific features [44]. These antibodies target the MLD in GP. Both antibodies are highly effective in mice (90%; Appendix A) and specific to EBOV only [44]. The MB-003 drug, which is described in Section 5.1, is based on antibodies 13F6-1-2, 13C6, and 6D8.

Takada et al. obtained murine monoclonal antibodies, which were specifically bound to EBOV GP [45]. VSV pseudotyped with EBOV GP was used as the immunization agent to induce the production of specific antibodies in the study [46]. Two of the three antibodies identified, namely 133/3.16 and 226/8.1, showed neutralizing activity and provided protection in mice upon passive immunization a day before virus initiation and two days after infection. In addition, 226/8.1 binds to the viral glycoprotein within GP1, while 133/3.16 binds to GP2 (Appendix A). A study of the ability of these antibodies to protect against a lethal virus dose provided ambiguous results. The administration of both antibodies in mice turned out to be effective; however, the passive immunization of guinea pigs with 133/3.16 and 226/8.1 did not prevent death in animals [47]. However, the result seems to be not ambiguous at all, considering the fact that the monoclonal antibodies used in the study are of murine origin and, therefore, heterologous to guinea pigs.

Chimeric antibodies ch133 and ch226 were obtained based on 133/3.16 and 226/8.1. The murine immunoglobulin domain in these chimeric antibodies was replaced with the corresponding constant domain of human immunoglobulin. The passive administration of the combination (ch133 + ch226) in monkeys a day before infection with 10^3^ pfu made it possible to delay death in one animal and prevent death in another monkey while the third animal died [48].

Qiu et al. used the hybridoma technology and obtained eight murine antibodies binding to the EBOV glycoprotein [49]. One of the antibodies binds to secreted GP (sGP; 1–295 aa), three antibodies interact with the MLD (333–458 aa), three antibodies interact with the epitopes at the C-terminal region of the GP1 domain (296–501 aa), and one antibody binds to the full-length GP. Previously, the same research team tried to obtain antibodies against EBOV; however, these antibodies showed no therapeutic effectiveness [50]. In contrast to the first attempt, the antibodies produced afterwards were verified using mouse-adapted EBOV and turned out to be effective. Mice received a viral dose of 103 LD50; antibody therapy was initiated either a day before or a day after infection. The survival rate increased in all cases. However, only five antibodies (4G7, 5D2, 5E6, 7C9, and 7G4) provided complete protection [49].

A study of the effect of the time of antibody administration on survival in mice demonstrated that the optimal time is a day before and a day after infection. Furthermore, 4G7 demonstrated partial protection if administered on day 3 after infection. Despite the fact that monotherapy did not protect guinea pigs, the use of antibody cocktails provided complete survival. The combination (1H3 + 2G4 + 4G7) provided the best result. This cocktail was called ZMAb [51].

A detailed study of antibodies 1H3, 2G4, and 4G7 made it possible to better understand the mechanisms underlying their protective effect. For instance, it turned out that 2G4 and 4G7 share high similarity. Firstly, their light chains have similar amino acid sequences. Secondly, the epitopes of these antibodies overlap significantly. Meanwhile, the 2G4 epitope apparently contains only the GP2 region, while the 4G7 epitope has an additional GP1 fragment. The antibody 1H3 practically does not compete with 2G4 and 4G7 for binding. Taken together, these data indicate that the antibody epitope differs significantly from those of 2G4 and 4G7 (Appendix A). It is worth noting that the protection mechanism of 1H3 action differs from those of 2G4 and 4G7: while the latter demonstrates a conventional neutralization pattern, 1H3 practically does not neutralize the virus, although it does exert some activity in in vivo experiments. This led us to the hypotheses on sGP recruitment by this antibody and the involvement of the mechanisms of antibody-dependent cellular cytotoxicity (ADCC) in this process [52]. Studies [43,53] describe in detail the results of epitope mapping for antibodies 1H3, 2G4, and 4G7.

In 2011, an antibody capable of neutralizing SUDV was obtained. A protocol similar to that described by Wilson et al. was used to produce this antibody [42]. An alphavirus replicon containing a GP-encoding sequence was used for immunization. After a series of immunizations in mice, the spleen was removed to produce hybridomas. A study of the structure of the 16F6–GP SUDV antibody complex made it possible to identify the epitope this antibody binds to. Similarly to KZ52, 16F6 binds to the region within the GP1–GP2 interface (Appendix A). However, the contribution to the effect of KZ52 is 20% and 80% for GP1 and GP2, respectively, while the same values for 16F6 are 60% (GP1) and 40% (GP2). The similarity to KZ52 is not limited to the region of interaction; both antibodies use similar mechanisms to combat the virus: they inhibit viral GP rearrangement, which is required for virus penetration into the cell [54].

Chen et al. [55] obtained humanized antibody variants based on 16F6. For this, a synthetic human YADS1 scaffold was used. This scaffold served as the basis for the secondary phage library of antibodies; 17 antibodies with different neutralizing activity against SUDV pseudoviruses were selected from this library. Two antibodies of this panel, namely E10 and F4, which have neutralizing properties and a good combination of physical and chemical properties, were tested in mice. Both antibodies had activity similar to that of 16F6 and protected 80–100% of animals from 10^3^ pfu of the virus. The neutralization assay with and without complement proteins demonstrated that the mechanism underlying the effect of these antibodies is related to neither the opsonization nor activation of the complement system.

Bispecific antibodies that are able to cross-link to EBOV and SUDV were developed on the basis of 16F6 (humanized variants E10 and F4) and KZ52 [56].

Using the approaches previously developed for the search and generation of broadly neutralizing antibodies, including those against HIV-1, Bornholdt et al. obtained several hundred antibodies based on the set of antibodies from a convalescent donor. Such a large experimental scale made it possible to not only obtain novel antibody variants with excellent characteristics but also elucidate the general pattern of epitope specificity against EBOV GP in an organism [57]. Apparently, these data do not allow one to make generalizations since similar data are required from a larger number of survivors. However, they also allow for a deeper glance into molecular aspects of the humoral response to filovirus infection.

Competitive data analysis divides all antibodies into several groups. The main ones among them are the following groups: GC-interacting antibodies (13C6 competitors), the group of antibodies binding to the region at the GP1–GP2 interface (KZ52 competitors), and antibodies interacting with the GP “base”. Combined together, these three groups of antibodies comprise 86% of all EBOV GPs.

An analysis of the neutralizing properties of these antibodies by the plaque reduction neutralization assay showed that 77% and 63% of them inhibited 50–80% of viral particles (PRNT50 and PRNT80), respectively, at a concentration of ≤50 µg/mL. However, another fact draws more attention: some of the antibodies can neutralize viruses at much lower concentrations compared to first-generation antibodies (for instance, PRNT80 is 0.625 µg/mL and ≤0.03 µg/mL for KZ52 and ADI-15848, respectively). A study of the therapeutic characteristics of representatives of each group showed that even their single administration (monotherapy) results in 100% survival [57].

Zhang et al. used a similar approach for the production of highly effective antibodies. However, the antibody pool was obtained artificially by immunizing primates with recombinant EBOV GP, lacking the transmembrane domain and MLD eight times. Of the five antibodies interacting with GP, only three (Q206, Q314, and Q411) neutralized the virus. The half maximal inhibitory concentration (IC50) of these antibodies is similar to the IC50 of G7, 2G4 [51], and 133/3.16 [46] but higher than that of new-generation antibodies ADI-15742 [58] and M318 [59]. According to the electron microscopy data, all three antibodies interact with GC. However, a more precise mapping of epitopes allows one to assume that two antibodies, namely Q206 and Q411, share an epitope, which, similar to the GC sequence, also contains a fragment of the central GP region. The Q314 epitope overlaps significantly with the 13C6 epitope (Appendix A). Antibodies Q206, Q314, and Q411 did not exert 100% protectivity, although they did enhance survival significantly [60] (Appendix A).

Studies [61,62] describe broadly neutralizing antibodies rEBOV-520 and rEBOV-548. Both antibodies contain an epitope binding to the GC of the EBOV and Bundibugyo virus (BDBV). Antibody rEBOV-548 neutralizes viruses and activates the effector functions. Antibody rEBOV-520 tightly interacts with the main regions in GP and GC (Appendix A). When used together, rEBOV-520 and rEBOV-548 can completely protect NHPs from death (Appendix A).

Based on the pool of antibodies from a Kikwit outbreak survivor (DRC, 1995), Corti et al. generated highly effective monoclonal antibodies mAb100 and mAb114 [63]. These two antibodies, as well as an additional 38 variants, were obtained after the sorting and immortalization of B cells. However, these were the only variants selected for the study since they could neutralize Ebola pseudoviruses. The combination (mAb100 + mAb114) in rhesus macaques demonstrated high therapeutic potential (Appendix A).

In the case of mAb114, 100% protection was observed even if a delay in administration occurred for as long as five days [63]. An explanation for this can be found by analyzing the results of the study on the complex of mAb114 and EBOV GP. It turned out that this antibody, despite the fact that it competes with the 13C6 antigen in its attempts to bind to GC, binds to the core sequence in GP1 (LEIKKPDGS epitope) to a greater extent and to GC to a lesser extent (Appendix A). This is the reason why mAb114 blocks a potential binding region between GP and NPC1, which is currently considered the main target for viral GP on the cell surface [64,65]. The mAb114-GP complex does not degrade at lower pH values and by cathepsin proteolysis. The lifespan of mAb114 when used as a preventive drug is 24.2 days, according to [66].

In 2020, EBANGA, which is based on mAb114, was registered in the USA to treat EBOV infection in adults and children, including infants [65] (see Section 5.5 and Appendix A).

Antibody mAb100 also has a number of characteristics that provide superiority over its competitor, KZ52. The complex of this antibody with GP has also decreased sensitivity to low pH values and structure changes under the effect of cathepsin L. Furthermore, unlike KZ52, the interaction of which is limited to the promoter only, mAb100 captures the IFL of the neighboring promoter when interacting with the GP1–GP2 interface region, thus interfering with the conformational rearrangements of GP required for virus penetration into the cell [67] (Appendix A).

The study in ref. [68] describes the production of antibodies against EBOV GP via the immunization of VelocImmune mice with a DNA vaccine encoding EBOV GP and the purified EBOV GP protein. As a result, the authors selected three antibodies containing non-overlapping epitopes on the GP surface: (1) REGN3470 (atoltivimab), which binds to GC and can activate the effector function; (2) REGN3471 (odesivimab), a neutralizing antibody overlapping with residues at the top of GP1, which interacts in a manner similar to mAb114 and activates the effector function; and (3) REGN3479 (maftivimab), which binds to the region at the GP1–GP2 interface and prevents IFL internalization into the endosome [68,69,70] (Appendix A). A cocktail of the three antibodies at a ratio of 1:1:1 has been registered under the brand name Inmazeb. Its effectiveness in combating EVD has been confirmed; more information on this drug is provided in Section 5.6 and Appendix A.

## 4. Cross-Reactive Antibodies

The recent efforts of research groups have focused on the production of antibodies reactive to a series of filoviruses. This is because not only EBOV but also BDBV, SUDV, and MARV pose a threat. It is impossible to predict in advance which pathogen will cause the next epidemic outbreak, while the development of more than one drug is less reasonable than creating a universal one, especially considering that the discovery of the phenomenon of broadly neutralizing antibodies [71,72] indicates the fundamental possibility of developing these antibodies. Technically, the production of bispecific antibodies by Frei et al. [56] and Wec et al. [73] can also be considered as one of these studies. However, works on the search for “true” broadly neutralizing antibodies should be considered first.

In order to obtain antibodies cross-reacting with filoviruses, Marceau et al. used the staggered immunization of mice with recombinant VSV pseudotyped with GP from EBOV, SUDV, and MARV. As a result, two monoclonal antibodies were obtained: S9 and M4. M4 showed neutralizing activity against MARV, while S9 neutralized only EBOV (Appendix A). Both antibodies protected mice and guinea pigs from the lethal virus dose [74]. Further efforts of the research team to find cross-reactive antibodies did not lead to any success. However, they found four antibodies cross-reacting with EBOV, SUDV, BDBV, the Reston virus (RESTV), and the Tai Forest virus (TAFV) [75].

The ability to neutralize not only EBOV but also SUDV, RESTV, and TAFV was shown for antibody #3327, obtained by P. Reinard and V. Volchkov using hybridoma technology [76] (Appendix A). This discovery is quite unexpected since, according to the analysis of resistant mutants, the amino acid residue at position 508 of GP plays a key role in binding, while the same region within GP2 is quite variable. The Gln508 mutation in GP prevents the recruitment of a series of neutralizing antibodies, including KZ52, c2G4, and c2G7. Pseudotyped VSV particles carrying this mutation are effectively neutralized by #3327, which indicates an important difference between these neutralizing antibodies (KZ52, c2G4, and c2G7) and #3327.

In order to obtain cross-reactive antibodies, Holtsberg et al. developed an original strategy to immunize mice with a combination of recombinant GPs from EBOV, SUDV, and MARV. After a series of boosting rounds, splenocytes were harvested and further used for the generation of monoclonal antibodies. Five out of more than one thousand antibodies obtained (m16G8, m8C4, m17C6, m4B8, and m21D10) had the ability to cross-react, while only one antibody, namely m21D10, interacted with GP of EBOV, SUDV, BDBV, RESTV, and MARV. The epitope of this antibody is located in the RBS region and conserved among flaviviruses. An analysis of the ability of antibodies m16G8, m8C4, m17C6, m4B8, and m21D10 to neutralize VSV pseudotyped with GP from EBOV and SUDV demonstrated that only one antibody, namely m8C4, exerted a significant effect (IC50 is 1.5 µg/mL for EBOV and 0.75 µg/mL for SUDV) Antibody m4B8 as a monotherapy and m16G8 in combination with m8C4 exerted a protective effect with 100% and 80% success rates, respectively. Antibody m21D10 did not show any protective properties [77] (Appendix A).

A similar approach was used by Z. Keck et al. [78] for the induction of cross-reactive antibodies but in animals other than macaques. A combination of three recombinant GPSs from EBOV, SUDV, and MARV lacking MLD was used for priming immunization, while a combination of VLPs EBOV, SUDV, and MARV was used for boosting. Next, the authors used two alternative approaches instead of hybridomas to obtain individual antibody variants: yeast display and sorting. Firstly, a yeast antibody library was generated. A series of chimeric antibody variants selected from this library (FVM01p, FVM02p, FVM04, FVM09, FVM13, and FVM20) were indeed bound to GP in EBOV, SUDV, BDBV, and RESTV, while two antibodies (FVM02p and FVM04) bound to MARV GP, with only one antibody, FVM04, showing neutralizing activity against both EBOV and SUDV. This antibody, when used as a monotherapy, provided 100% survival in mice infected with 10^3^ pfu EBOV. The remaining antibodies had lower protective efficacy. At the same time, the use of antibody cocktails made it possible to increase the survival. For instance, the use of (FVM02p + FVM09) increased the survival to 100%.

The study of structural features of FVM04 interaction with filovirus GP elucidates the origin of its effectiveness. According to cryo-electron microscopy and alanine scanning mutagenesis data, this antibody can be conditionally combined into one group with 13C6 and 1H3, since all of them interact with GC. However, there are fundamental differences. Similarly to the mAb114 epitope, the FVM04 epitope contains a GP region responsible for the interaction with NPC1 (Appendix A). Such a precise interaction allows FVM04 to effectively inhibit virus penetration into the cell, while GP proteolysis by cathepsin only enhances the inhibitory effect. This region within FVM04, which plays an important role in interaction between GP and NPC1 is highly conserved; therefore, this antibody not only has an extremely wide reactivity (against EBOV, SUDV, BDBV, and RESTV) [72] but can also potentially neutralize novel EBOV variants with a modified GP structure, provided that the interaction region within FVM04 remains unchanged since it plays a crucial role [79]. One can hope that the region found by Howell et al. [79] will turn out to be the Achilles’ heel of EBOV similar to the CD4 binding site in HIV-1 [80].

The second approach, including the sorting of specific B cells from immune macaques, also made it possible to find an effective antibody variant. Of the 12 clones selected by the authors of this study, only CA45 turned out to be able to neutralize EBOV, SUDV, BDBV, and RESTV. Since the epitope in this antibody is located at the GP1–GP2 interface, partially overlapping with the FVM04 epitope, it is reasonable to use a complex of these antibodies as a therapeutic agent. The correctness of this assumption was demonstrated in mouse and guinea pig models: the single administration of (CA45 + FVM04) on day 3 after infection provided the 100% survival of both species [81].

CA45 activity against Filoviridae family representatives is impressive, especially considering the fact that first-generation antibodies interacting with this region (KZ52, 16F6, 4G7, and 2G4) have a narrow specificity range. This phenomenon can be explained by the presence of a unique epitope containing an IFL fragment, a region within GP2, which is highly conserved among ebolavirus representatives (the key amino acids involved in the interaction with CA45, Y517, G546, and N550 are conserved), and a conserved region at the N-terminus of GP1 (amino acid residue R64) [81] (Appendix A).

Other authors [82] studied the cocktail of antibodies FVM04, CA45, and MR191 for the protection of NHPs from EBOV, SUDV, and MARV. This cocktail has a broad neutralizing activity due to the inhibition of the key epitopes at the top and base of GP. In particular, it showed 100% effectiveness in MARV-infected primates.

An antibody capable of neutralizing not only EBOV (1976 and 2014 outbreaks) but also SUDV, BDBV, RESV, and TAFV was obtained by Furuyama et al. by immunizing mice with EBOC and SUDV VLPs selected using the hybridoma technology. An analysis of resistant EBOV GP variants demonstrated that, like CA45, 6D6 also targets IFL. The single administration of 100 µg of the antibody provided protection from a lethal dose of EBOV-adapted strains in mice [83].

Cross-neutralizing antibodies were obtained when studying the humoral response after BDBV infection. Using data on serum cross-reactivity against EBOV and SUDV, the authors performed the immortalization of B cells and analyzed 90 individual monoclonal antibodies. A total of 57 out of 90 antibodies turned out to be capable of binding to more than one heterologous GP. Some of the antibodies were able to not only bind but also exert a neutralizing effect as well. A total of 31 out of 90 antibodies were neutralizing (against BDBV). Most antibodies neutralized either BDBV, BDBV, or EBOV. However, two antibodies, namely BDBV223 and BDBV289, also neutralized SUDV. These antibodies can protect mice and guinea pigs from a lethal dose of EBOV (Appendix A) [84].

Taking into account the fact that these antibodies bind to different regions within viral GP (BDBV223 interacts with GP2, while BDBV289 binds to GC), it would be reasonable to use a combination of these antibodies to treat the EBOV infection. A more successful result was obtained when using this combination compared to monotherapy. It is also interesting to note that a large percentage of neutralizing antibodies obtained in this study were targeted at GC, which is considered an unfavorable site for effective antibodies due to its sensitivity to cleavage by cathepsin [84].

The screening of the variety of antibodies obtained from a survivor of the 2014 Ebola virus outbreak [57] against rVSV BDBV, SUDV, and EBOV made it possible to select cross-reactive variants, with some of them capable of broadly neutralizing. Neutralizing antibodies were found among all groups (binding to the base, GP1–GP2 interface, and GC) previously identified by the authors. However, antibodies to the GP1–GP2 interface, which shares a target with KZ52 (as well as 4G7, 2G4, and CA45), isolated almost 20 years ago, are characterized by broad reactivity and effectiveness. This is another evidence in favor of the fact that this GP region in EBOV is one of the most promising targets for the creation of immunotherapy drugs.

The same laboratory later reported two broadly neutralizing antibodies, namely ADI-15878 and ADI-23774, which were used as the basis for the antibody cocktail MBP134 and capable of neutralizing all EBOV species [85]. ADI-15878 and ADI-23774 target unique non-overlapping GP epitopes and neutralize both the extracellular and endosomally cleaved GP. MBP134 provides 100% protection in ferrets and NHPs when challenged with 10^3^ pfu. Treatment was carried out using a single dose of 25 mg/kg on day 5 after infection (Appendix A).

It has been previously noted that, despite the fact that antibodies KZ52, 4G7, and 2G4 compete for binding to the same region, the molecular mechanisms of their action differ. This is manifested in not only different affinities but also different productivity of these antibodies [86]. Despite the fact that antibodies ADI-15946, ADI-15878, and ADI-15742 compete with KZ52 for binding (Appendix A), the features of their interaction with GP allow us to classify them as a separate group. While ADI-15946 is similar to KZ52 (as well as 4G7 and 2G4) in the mechanism of virus inhibition, it impedes the cathepsin-mediated processing of GP in the endosome. Furthermore, unlike KZ52, ADI-15946 acts at late stages of the process, while ADI-15878 and ADI-15742 have different mechanisms of action.

The antibody ADI-15878/15742 can inhibit the last stage of virus penetration, which takes place after the enzymatic cleavage of MLD and GC. The interaction of antibodies with “naked” GP (after the elimination of variables MLD and GC), apparently, explains such a broad neutralization range. It is also important to note that while studying the possibility of producing antibodies similar to ADI-15878/15742 in the course of natural infection, the authors demonstrate the absence of fundamental barriers [52]. A relatively short CDR-H3 loop (15 aa long) with a small number of somatic mutations (6%) indicates the short time period required for the generation of these antibodies, which makes them different from such antibodies as broadly neutralizing antibodies against HIV-1 [71].

The study [87] demonstrated that ADI-15878 binds to conserved regions in HR1 and IFL domains of EBOV, SUDV, and BDBV. The use of this antibody resulted in 33–50% survival in infected guinea pigs.

Bioengineering technologies make it possible to create protein molecules that are absent in living nature. One of these technologies associated with immunotherapy drugs is the use of bispecific antibodies. The use of this technology, together with variable KZ52 domains E10 and F4, yielded a number of constructs that combined two features in one molecule. The efficiency of this approach was demonstrated by conducting the rVSV-GP (EBOV and SUDV) neutralization reaction in vitro. Two proteins with improved characteristics, scKZ52-F4HCC and scKZ52-F4LCN, were tested in animal models. The best results were shown for scKZ52-F4HCC, which provided 100% survival in mice (Appendix A) [56].

An even more sophisticated strategy for creating an engineered drug was proposed by Wec et al. [73]. In the study of antibodies against filoviruses, a number of authors obtained antibodies targeted at highly conserved regions within GP but showing no neutralization signs for different reasons. These antibodies included FVM09 (which interacts with the conserved region within GC; 283GEWAF289) and MR72 (which interacts with the conserved region at the top of GP and is responsible for interaction with NPC1) [78,88]. Variable fragments of these antibodies, as well as the antibody interacting with NPC1 (and capable of inhibiting access to this region), served as the fragments of the antibody-like protein able to internalize into the endosome together with the virus and then inhibit the subsequent stages of virus penetration into the cell.

Thus, the production of monoclonal antibodies against EBOV includes three stages. The first stage is associated with the first attempts to obtain antibodies targeted at EBOV. Most of the tools and data available to date were absent during this stage. The study of structural features of EBOV GP was often performed in parallel with the study of antibodies interacting with GP. Not all antibodies discovered at the first stage (KZ52, 13C6, 6D8, 13F6-1-2, 1H3, 2G4, 4G7, etc.) are suitable for the creation of an effective drug. Second-generation antibodies (EBOV-520, EBOV-548, mAb100, mAb114, etc.) seem to be more promising. This applies both to the mechanism of their action, which has been studied in detail, and their effectiveness. Third-generation antibodies (#3327, ADI-15878, ADI-23774, FVM04, CA45, etc.) and antibody-based recombinant immunotherapy drugs should become not only effective but also universal, capable of protecting against a wide spectrum of filoviruses. However, it should be noted that these antibodies are developed to combat mainly EBOVs as the main threat.

## 5. Drugs Based on Monoclonal Antibodies

### 5.1. MB-003

MB-003 was one of the first drugs based on monoclonal antibodies to treat EVD. This drug is based on three murine antibodies: 13C6, 13F6-1-2, and 14G7. In order to use antibodies 13F6-1-2 for therapy, their immunogenicity had to be decreased, and they had to be humanized. As a result, the h-13F6 antibody variant was obtained. Two systems were used for preparative antibody production: mammalian CHO cells and transient expression in plant cells (magnICON) [89,90]. The best results were achieved using plant cells. Carbohydrate chain composition was the key factor. The absence of a fucose residue made antibodies more active (median effective dose (ED50) = 3 μg versus ED50 = 11 μg) for antibodies produced in CHO cells [89].

Considering the previous experience of evaluating the protective effect, two other antibodies from this preparation were also produced in two systems: CHO and plant cells. Antibodies were then tested in NHPs. The results of this analysis turned out to be rather ambiguous. In the first experiment, two (100%) NHPs treated with MB-003 survived, while one (100%) control NHP died. In the second experiment, one (20%) animal receiving MB-003 died, and four (80%) animals receiving MB-003 survived. One (100%) control animal survived as well. Finally, two (34%) experimental animals receiving MB-003 died, while four (67) survived in the third experiment (all control animals were dead). It is also important to note that the preparation was administered three or four times (in different experiments) [91]. In order to elucidate the reasons for this, a virus population originating from deceased animals was studied. In particular, a thorough analysis of nucleotide substitutions in the genome was performed. It turned out that isolates obtained from one of the animal samples were not neutralized by MB-003, while single-nucleotide polymorphism in the genome results in five significant amino acid substitutions: K272N, T283A, D397G, Q406R, and K395R/G/E [92]. According to [43], four of these substitutions are crucial for the interaction between MB-003 and EBOV GP. On the one hand, this result sheds light on the animal death observed in experiments. On the other hand, it raises questions about the necessity of selecting the components for monoclonal antibody-based drugs more thoroughly. The results of the following experiment using MB-003 can hardly be considered successful. This experiment was aimed at analyzing the therapeutic ability of the drug upon its administration after the manifestation of detectable disease symptoms (imitation of real disease condition), viremia (RT-PCR), and temperature growth. Only three (43%) out of seven experimental animals survived. The average period of MB-003 administration was three days after infection (Appendix A) [93].

### 5.2. ZMAb

The ZMAb drug is very similar to the above one in the history of its creation and ideology. The first successful use of the combination of antibodies 1H3, 2G4, and 4G7 in rodents [51] became the impetus for testing ZMAb in NHPs. Two experiments were conducted in two groups of cynomolgus monkeys. In the first group, the antibody preparation was administered in animals a day after infection, while the second group received the drug two days after infection. All animals in the first group survived, while the survival rate in the second group was only 50%. This result became a significant event since the previous attempts to treat this infection in monkeys using monoclonal antibodies failed (antibodies KZ52, ch133 + ch226, and the MB-003 drug). The drug dose for ZMAb was two times lower (25 mg/kg) than that of KZ52 (50 mg/kg) and approximately two times higher than that of ch133 + ch226 (~12,5 mg/kg). The frequency of drug administration, which equaled three for ZMAb (1,4,7 or 2,5,8 days after infection), was the same for ch133 + ch226 but higher than that of KZ52 (two injections) [94]. The next step included the assessment of the possibility of the immune system of NHPs who survived due to passive immunization with ZMAb to combat the secondary administration of the virus after 10 (group 1) and 13 (group 2) weeks. All six monkeys in the first group survived, while only four out of six animals were saved in the second group. A study of animal immunity showed that the passive administration of antibodies did not interfere with the development of protective immunity. The authors suggest that the death of the two animals in the second group was due to the longer period after the first viral infection, as well as the features of the immunization strategy and the animal immune system [95] (Appendix A).

The protective effect of ZMAb was also studied in combination with DEF201 (replication-defective recombinant human adenovirus of serotype 5 encoding the interferon-alpha sequence). DEF201 administration in guinea pigs a day after infection resulted in 100% survival in the case of a five-day delay in ZMAb administration when using a dose of 5 mg per animal and a seven-day delay in treatment when using an antibody dose of 10 mg per animal [96]. A 100% survival rate was achieved in rhesus macaques in a similar experiment upon the combined administration of DEF201 and ZMAb (50 mg/kg) on day three after infection (10^3^ pfu). The survival rate in cynomolgus monkeys was 75% when using a similar administration strategy [97].

### 5.3. Zmapp

In 2014, the drug named Zmapp was created based on MB-003 and ZMAb components. A series of experiments were conducted in guinea pigs and NHPs using individual antibodies comprising MB-003 and ZMAb in order to select the most optimal composition. The Zmapp1 cocktail (later called Zmapp), which includes antibodies c13C6 (MB-003), c2G4 (ZMAb), and c4G7 (ZMAb), demonstrated the best result. Zmapp differs from ZMAb in a c1H3 to c13C6 substitution. Both antibodies (c1H3 and c13C6) interact with GC. Furthermore, as mentioned above, the epitopes of these antibodies overlap partially, although they are incompletely identical [41,43]. Nevertheless, this substitution made it possible to enhance the protective effect of the drug, as demonstrated using three therapeutic strategies. A total of 50 mg/kg of the drug was injected thrice at a three-day interval. The first, second, and third strategies implied drug administration three, four, and five days after infection, respectively. Zmapp use in monkeys protected animals from death (viral dose of 628 pfu) even in cases when the treatment was started five days after infection [98] (Appendix A). An increase in drug effectiveness after the c1H3 to c13C6 substitution could be explained by a change in the antibody affinity: the affinity of c13C6 to GP is 10 times higher than that of c1H3 [43].

The Ebola outbreak in 2013–2016 demanded extraordinary efforts in order to obtain a significant amount of drug doses. The use of plant-based technologies does not allow one to produce the required drug dose amounts. Therefore, a decision was made to use variants generated in CHO cells. This transformation affected the technological process, and not all antibodies comprising the cocktail retained the features required for their effective production. For instance, 13C6FR1 (a 13C6 variant with a modified FR1 sequence) was subjected to aggregation during its production in CHO cells and demonstrated reduced thermal stability. The substitution of the amino acid at position 148 to Lys improved the physical and chemical characteristics of this antibody [99].

The clinical trial of Zmapp was launched by volunteers at the beginning of March 2015. A total of 71 individuals participated in the study. Of these, 35 people received maintenance therapy, while 36 individuals had standard treatment with Zmapp. The results of Zmapp administration were as follows: 8 (22%) and 13 individuals (37%) died in the target and control groups, respectively. This result can be considered significant; however, the sample size does not allow accurate conclusions to be drawn [100].

The only large clinical trial was conducted during the EBOV outbreak in DRC in 2018. The trial was aimed at the evaluation of experimental drugs against EVD in patients in the field conditions. Different disease stages were confirmed by RT–PCR. This study compared the following drugs: ZMapp, mAb114, REGN-EB3, and remdesivir (remdesivir is a monophosphate nucleoside analog (GS-441524), which acts as a viral RNA-dependent RNA polymerase inhibitor, targeting the viral genome replication process). Remdesivir has previously demonstrated anti-viral activity against EBOV and MARV in vitro and in vivo in animal models. More recently, remdesivir has demonstrated efficacy against coronaviruses (SARS-CoV, MERS-Co-V, SARS-CoV-2), paramyxoviruses, pneumoviruses, and is the first FDA-approved anti-viral agent. More information about remdesivir can be found at [101]). Drugs were injected in 681 EBOV-infected patients at a ratio of 1:1:1:1. In the study [102], patients in the ZMapp group received a dose of 50 mg per kilogram of body weight every third day starting on day 1 (for a total of three doses). Patients in the remdesivir group received a loading dose on day 1 (200 mg in adults and weight-adjusted in children) and then a daily maintenance dose (100 mg in adults) starting on day 2 and continuing for 9–13 days, depending on the viral load. Patients in the MAb114 group received a dose of 50 mg per kilogram administered as a single infusion on day 1. Patients in the REGN-EB3 group received a dose of 150 mg per kilogram administered as a single infusion on day 1. As a result, ZMapp and remdesivir had no statistically significant effect: EVD case fatality rates were 49.7 and 53% for ZMapp and remdesivir, respectively [100,103]. No difference was found between Zmapp and remdesivir use and standard treatment. Similarly to mAb114 and REGN-EB3, the fatality rates were 35.1 and 33.5%, respectively [104].

### 5.4. MIL77

The authors [105] further tried to improve the (c13C6 + c2G4 + c4G7) antibody cocktail. The reason for this was their desire to improve the glycosylation profile as well as insignificant levels of 4G7 synthesis. The improved c2G4 variant named MIL77-1 contains five amino acid substitutions compared to the original one, while MIL77-2 (a c4G7 variant) and MIL77-3 (a c13C6 variant) have three and twenty substitutions, respectively. All substitutions were introduced in the constant antibody region and did not affect antigen-recognizing regions. As a result, the glycosylation profile was improved, which enhanced the affinity of the antibodies to the Fcγ IIIa receptor; however, the yield of MIL77-2 antibodies was lower than those of MIL77-1 and MIL77-3. Thus, considering the fact that c4G7 and c2G4 interact with the overlapping epitope, a decision was made to test two combinations of antibodies: MIL77-1 and MIL77-3. Experiments in guinea pigs demonstrated that the resulting drug is at least as protective as ZMapp, while drug studies in NHPs showed an even higher protective effect compared to ZMapp [105]. Clinical trials of MIL77 were carried out by the owner company, Mapp Biopharmaceutical, and turned out to be unsuccessful, according to [106].

In [107], the authors simulated EBOV infection in vaccinated (with rVSV-ZEBOV) and unvaccinated rhesus macaques. Animals received MIL77 three days after infection with a lethal virus dose. As a result, vaccinated primates and primates treated with MIL77 demonstrated no clinical signs of disease and survived (Appendix A). On the contrary, animals receiving either vaccination or mAb therapy became ill with decreased survival rates. Based on these data, we can assume that MIL77 can be used in combination with other drugs against EBOV.

### 5.5. Ebanga (Ansuvimab)

As stated above, Ebanga (or Ansuvimab) is the EBOV-neutralizing monoclonal antibody mAb114 (EC50 0.06–0.15 µg/mL); the maximal ADCC is achieved when using an Ansuvimab concentration of 0.03 µg/mL [65]. The antibody is produced in CHO cells [101]. More information on the characteristics and adverse effects of the drug can be found in [59].

Ebanga trials during the EBOV outbreak in DRC in 2018 showed that the drug has a protective therapeutic effect and decreases mortality by up to 35.1% among all individuals infected with EBOV [103]. The fatality rate in the control group of infected individuals without treatment was 49.4% [104]. A total of 50 mg/kg of Ansuvimab was used to treat patients with a confirmed EVD diagnosis of any age (174 patients) in addition to standard therapy. Porgaviximab at a dose of 50 mg/kg on days 1, 3, and 7 was used as an active control (169 patients). The average time before receiving the first negative PCR result for EBOV was 16 days after Ansuvimab administration and 27 days after treatment with Porgaviximab [65].

Based on the results of the experiments, the researchers concluded that the recommended dosage of Ansuvimab was 50 mg/kg via intravenous infusion over 60 min [65].

### 5.6. Inmazeb (REGN-EB3)

Inmazeb is a combination of three antibodies: REGN3470 (atoltivimab), REGN3479 (maftivimab), and REGN3471 (odesivimab) at a ratio of 1:1:1. Antibodies are produced in low-fucose CHO cells [108]. The drug was registered in the USA in October 2020 (Figure 3).

Inmazeb trials started to study its effectiveness in NHPs. A three-stage experiment confirmed Inmazeb’s effectiveness in primates; the single administration of 150 mg/kg of an antibody cocktail was selected for experiments in humans.

The first randomized, double-blind, placebo-controlled study in humans was carried out in the USA. As a result, six out of 18 participants in the experimental group developed adverse effects of mild-to-moderate severity. Headache was the most common adverse event after treatment. There were no deaths or serious events that could have led to study discontinuation [64]. More details on the characteristics and side effects of the drug can be found in [109].

During the EBOV outbreak in DRC in 2018, REGN-EB3 demonstrated a protective therapeutic effect; the fatality rate in the group of patients receiving the drug was 17% lower than that of the control group [103,104].

After having conducted all trials and registered the drug, its recommended dosage was 50 mg/kg [109].

A detailed study [104], which presents a systematic review and meta-analysis, provides the following conclusions on mAb114 and REGN-EB3:In contrast to remdesivir and Zmapp, mAb114 and REGN-EB3 showed high effectiveness after single administration;There was no statistically significant difference between the effectiveness of mAb114 and REGN-EB3;No adverse effects were observed in patients in response to mAb114 and REGN-EB3 compared to standard treatment.

## 6. Conclusions

Ebola virus, one of the members of the filovirus group, with a fatality rate of up to 90%, is possibly one of the deadliest viruses in human history. Since its discovery, significant efforts have been made to develop methods for the prevention and treatment of infections caused by this virus. Both preventive and therapeutic research have progressed in parallel, and while effective vaccines have now been developed and are available, the same cannot be said for therapeutic drugs.

Historically, the first approach to treating the EVD is the use of serum antibodies. Since there were no survivors who could serve as a source of immune serum, the obvious step was to attempt the use of heterologous (equine, goat, sheep) polyclonal immunoglobulin preparations. These attempts can only be partially considered successful, but they demonstrated a crucial point: immunoglobulins can, in principle, form the basis of therapeutic drugs. Despite the characteristics of the virus, particularly its rapid infection progression, some promising results were obtained for serum-based treatments.

The next step in the development of immunotherapy for EVD was the creation of monoclonal antibodies. This direction relied on advances in the molecular biology of the virus and detailed studies of the structure of the primary target of neutralizing antibodies, the surface glycoprotein. Additionally, the general progress in methods for discovering and producing monoclonal antibodies drove this work. While first-generation antibodies, such as KZ52, 2G4, 1H3, and 4G7, had modest neutralizing activity against EBOV, later-discovered antibodies, such as mAb 114, REGN-3470, and ADI-15742, performed excellently during in vitro and in vivo experiments [110,111]. Given such impressive progress, one might have expected success in the clinical applications of these antibodies. However, clinical trials of Zmapp showed that, despite good results in primates, things work somewhat differently in humans. At the same time, the use of Ebanga and Inmazeb did indeed reduce mortality to an average of 34%. Does this mean that developing monoclonal antibody-based drugs to combat EVD is futile and should be abandoned? Not at all. First, it is important to understand the limitations imposed by the nature of the virus itself [112,113].

The Ebola virus causes hemorrhagic fever, an extremely rapid progression of the disease accompanied by liver and kidney dysfunction and extensive hemorrhaging. These characteristics impose significant limitations on how monoclonal antibody-based drugs can be used. Antibody-based drugs must be administered early in the infection, ideally immediately after exposure. Administering the drug at a stage where significant organ damage has already occurred may reduce viral load but will not counteract the damage to organs and tissues.

Given these limitations, can the situation be improved with existing capabilities? One approach could be the use of new antibody cocktails that interact with the maximum number of vulnerable regions in the virus, complementing each other without competition. Even weakly neutralizing or non-neutralizing antibodies, when combined, can provide neutralization and protection through synergistic effects [49,84]. Another approach could be the use of small-molecule inhibitors of viral proteins. For example, the use of remdesivir [114] appears to be a promising strategy, as this drug and monoclonal antibodies target fundamentally different viral components. A synergistic effect can be expected when they are used together.

## Figures and Tables

**Figure 1 antibodies-14-00022-f001:**
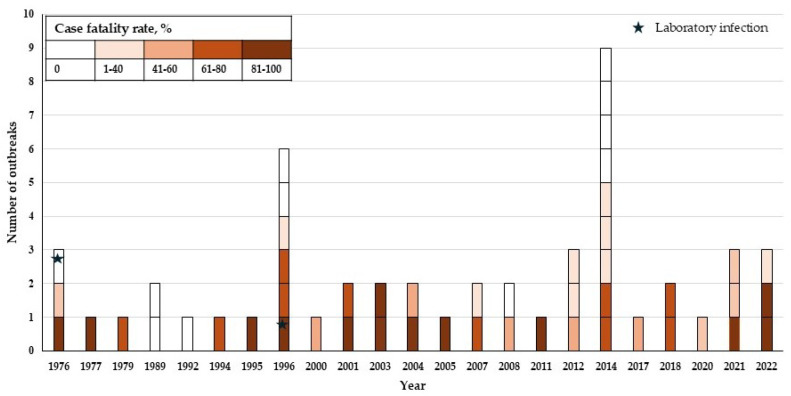
Number of Ebola virus disease (EVD) outbreaks. Outbreaks with a zero-case fatality rate are caused by the Reston virus and Tai Forest virus. The chart is based on the data of the US Centers for Disease Control and Prevention [https://www.cdc.gov/ebola/outbreaks/index.html, accessed on 28 November 2024]. An outbreak was defined as any reported case of one type of ebolavirus.

**Figure 2 antibodies-14-00022-f002:**
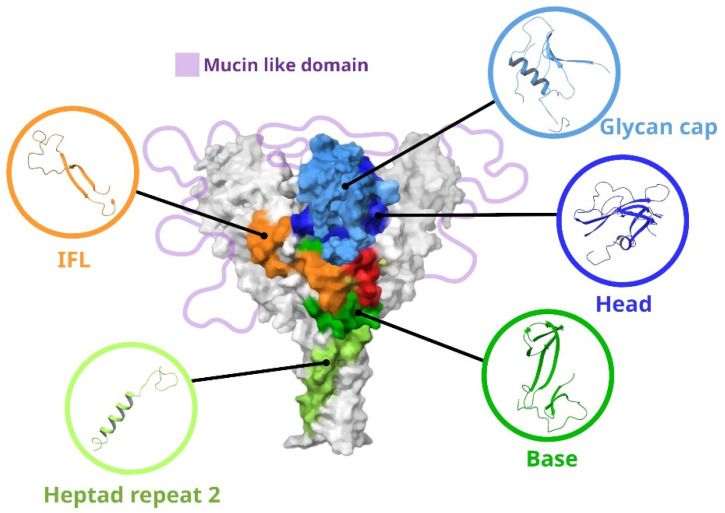
Main regions of vulnerability of ebolavirus GPs. IFL—internal fusion loop; Heptad repeat 2—region of Heptad repeat GP2; base—region base GP1; head—region head GP1; glycan cap—region including glycan cap GP1.

**Figure 3 antibodies-14-00022-f003:**
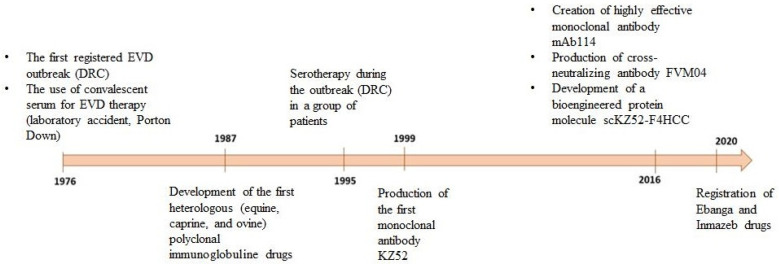
Chronological representation of drug development based on anti-EVD antibodies.

## Data Availability

No new data were created or analyzed in this study. Data sharing is not applicable to this article.

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
