# Peer review of "Treatment of Ebola Virus Disease: From Serotherapy to the Use of Monoclonal Antibodies"

_2073-4468, 2025, doi:10.3390/antib14010022_

Round 1

Reviewer 1 Report

Comments and Suggestions for Authors

I enjoyed the article. Congratulations to the authors. 

I have a comment about the references used in the article. Are authors included all published articles related to the review article's title here?

If yes please write about it in your article, If you selected some articles among a group, please mention the inclusion and exclusion criteria.   

Author Response

The team of authors would like to thank the reviewer for high evaluation of our work and making important comment.

I have a comment about the references used in the article. Are authors included all published articles related to the review article's title here?

If yes please write about it in your article, If you selected some articles among a group, please mention the inclusion and exclusion criteria.

 Thank you for your comment! We have added the following wording to the text in the introduction section.

In this review, we have attempted to cover as fully as possible the information concerning serum therapy and specific monoclonal antibodies since the discovery of the virus. The review was performed on all available experimental articles from 1977 to the present. In the case of several articles devoted to one monoclonal antibody, the article that most thoroughly described the structure and properties of the antibody was selected for the review.

Reviewer 2 Report

Comments and Suggestions for Authors

Authors keenly collated the EBV historical prelicnical and clinical research with great emphasis on antibodies therapies. It's a good recollection. Few points below.

Intro. What is the R0 for EBV. By the way, authors use EBOV and EBV, please seek uniformity.

It is missing a brief intro and description of the EBV - structure, proteins, family etc. Without such, Ab targeting is not connected.

When listing the drugs available, authors do not clarify which ones are approved, experimental or failed.

Low-to-none mention on remdesivir and potential to combination therapy with REG antibodies. Not clear when mAb should be given if early, during or at acute phase.

Authors should consider having a paragraph about drawbacks. For instance, why all the fuss on REG mAb antibodies during COVID19 and yet no is using them anymore for therapy? Authors should discuss about EBV evolution in this context.

Author Response

The team of authors would like to thank the reviewer for evaluating our work and making important comments.

  1. What is the R0 for EBV.

Thank you for your comment.

Typically, the basic reproduction number R0 for EBOV ranges from 1 to 2. It should be emphasized that R0 for EBOV varies depending on several factors, including the specific outbreak, the strain of the virus, and the effectiveness of public health interventions. For example, in the 2014-2016 West African EVD outbreak, the estimated R0 was around 1.5 to 2.5 before interventions were implemented. After interventions R0 can be significantly reduced to below 1. These changes in R0 value further highlight the need to develop preventive measures and therapies against EBOV.

The number and regularity of EVD outbreaks, as well as high case fatality rate, are presented in Figure 1 of the review. We believe that these data are sufficient to understand the danger of EVD and the importance of developing preventive and therapeutic means.

Due to the large volume of the article and the array of information, the authors ask that information about R0 not be included in the article.

By the way, authors use EBOV and EBV, please seek uniformity.

In this article, we use the common abbreviations EBOV and EVD. The former is used to refer to the virus, and the latter to refer to the disease caused by the virus.

  1. It is missing a brief intro and description of the EBV - structure, proteins, family etc. Without such, Ab targeting is not connected.

Thank you for your comment. We have added to the review brief information about the structure of the viral particle and the functions of proteins.

Ebolaviruses are single-stranded negative-sense RNA viruses. The virion is formed by seven proteins: nucleoprotein NP, polymerase cofactor VP35, matrix protein VP40, glycoprotein GP, transcription factor VP30, nucleocapsid-associated protein VP24 and RNA-directed RNA polymerase L [2]. NP, VP30, L, VP35 together with RNA form a ribonucleoprotein complex necessary for transcription and replication of the genome [3]. VP40 is able to penetrate lipid bilayers and initiate membrane curvature, leading to the formation of viral particles. VP24 blocks cellular immunity, regulates the synthesis of viral RNA, and is involved in genome packaging [4]. GP plays a key role in virus penetration and pathogenesis.

Since neutralizing antibodies are targeted at ebolaviruses GP, we will consider its structure in more detail. The image of GP with the main targets for antibodies is shown in Figure 2. GP is a trimer consisting of heterodimers. Each heterodimer contains a GP1 subunit (approximately 470 amino acid residues) and a GP2 subunit (approximately 175 amino acid residues) [6]. The GP1 subunit is responsible for cell attachment and can be divided into three subdomains: base, head, and glycan cap (GC). GP1 also con-tains a mucin-like domain (MLD) in the C-terminus, a region with several conserved cysteine residues at the N-terminus, and a receptor binding site (RBS) [7]. The GP2 subunit is responsible for the fusion of the viral and host cell membranes. GP2 contains conserved cysteine residues, an internal fusion loop (IFL), Heptad repeats 1 and 2, an alpha-helical transmembrane domain, and a region homologous to the conserved im-munosuppressive motif found in oncogenic retroviruses [8]. Most epitopes are located at the GP1–GP2 interface, GC, GP "base" (Table S1).

Figure 2. Main regions of vulnerability of Ebolavirus GP. IFL – internal fusion loop; Heptad repeat 2 – region of heptad repeat GP2; Base – region base GP1; Head – region head GP1; Glycan cap – region including glycan cap GP1.

  1. When listing the drugs available, authors do not clarify which ones are approved, experimental or failed.

Thank you for your comment. We have added the relevant information to Table S3.

  1. Low-to-none mention on remdesivir and potential to combination therapy with REG antibodies.

Thank you for your comment. The following information has been added to paragraph 5.3:

Remdesivir is a monophosphate nucleoside analog (GS-441524), acts as a viral RNA-dependent RNA polymerase inhibitor, targeting the viral genome replication process. Remdesivir has previously demonstrated antiviral activity against EBOV and MARV in vitro and in vivo in animal models. More recently, remdesivir has demonstrated efficacy against coronaviruses (SARS-CoV, MERS-Co-V, SARS-CoV-2), paramyxoviruses, pneumoviruses, and is the first FDA-approved antiviral agent. More information about remdesivir can be found at [Aleem, A., Kothadia, J.P. Remdesivir. StatPearls [Internet]. 2022, Available from: https://www.ncbi.nlm.nih.gov/books/NBK563261/; Bakheit, A. H., Darwish, H., Darwish, I. A., Al-Ghusn, A. I. Remdesivir. Profiles Drug Subst. Excip. Relat. Methodol. 2023, 48, 71-108. DOI:10.1016/bs.podrm.2022.11.003].

The following information has been added into conclusion:

Another approach could be the use of small-molecule inhibitors of viral proteins. For example, the use of remdesivir appears to be a promising strategy, as this drug and monoclonal antibodies target fundamentally different viral components. A synergistic effect can be expected when they are used together.

  1. Not clear when mAb should be given if early, during or at acute phase.

Thank you for your comment. The following information has been added to paragraph 5.3:

In the study [97], patients in the ZMapp group received a dose of 50 mg per kilogram of body weight every third day starting on day 1 (for a total of three doses). Patients in the remdesivir group received a loading dose on day 1 (200 mg in adults and weight-adjusted in children), then a daily maintenance dose (100 mg in adults) starting on day 2 and continuing for 9–13 days depending on viral load. Patients in the MAb114 group received a dose of 50 mg per kilogram administered as a single infusion on day 1. Patients in the REGN-EB3 group received a dose of 150 mg per kilogram administered as a single infusion on day 1.

The following information has been added into conclusion:

The Ebola virus causes hemorrhagic fever, an extremely rapid progression of the disease accompanied by liver and kidney dysfunction and extensive hemorrhaging. These characteristics impose significant limitations on how monoclonal anti-body-based drugs can be used. Antibody-based drugs must be administered early in the infection, ideally immediately after exposure. Administering the drug at a stage when significant organ damage has already occurred may reduce viral load but will not counteract the damage to organs and tissues.

  1. Authors should consider having a paragraph about drawbacks. For instance, why all the fuss on REG mAb antibodies during COVID19 and yet no is using them anymore for therapy? Authors should discuss about EBV evolution in this context.

Thank you for your comment. А discussion of the disadvantages of monoclonal antibodies is provided in the conclusion of the review. Рarticularly, the following information has been added into conclusion:

Among the disadvantages of using antibodies as a therapy, it is worth noting that when using monotherapy with antibodies or a cocktail of antibodies with overlapping epitopes, there is always a risk of mutation even in conserved epitopes of the virus, which leads to the emergence of its new resistant variants. This is especially true for rapidly evolving viruses such as SARS-CoV-2. Mutations in EBOV occur much more slowly, but given the CFR level for EVD, they can be extremely dangerous. The creation of antibody cocktails targeting non-overlapping epitopes, such as Inmazeb, can reduce the risk of drug resistance of the virus [Rayaprolu, V. et al. Structure of the Inmazeb cocktail and resistance to Ebola virus escape. Cell Host Microbe. 2023, 31, 260-272 doi: 10.1016/j.chom.2023.01.002.].

Reviewer 3 Report

Comments and Suggestions for Authors

This is a very nicely written review that describes the history and evolution of antibody-based interventions against Filoviruses. The only critique of the review is a relatively short conclusions section that does not discuss the challenges and the path forward in greater detail.

Author Response

The team of authors would like to thank the reviewer for high evaluation of our work and making important comment.

The only critique of the review is a relatively short conclusions section that does not discuss the challenges and the path forward in greater detail.

Thank you for your comment! We have expanded the conclusions section, described in more detail the advantages and disadvantages of using monoclonal antibodies, existing problems and ways to solve them, as well as strategies for developing drugs based on monoclonal antibodies.

Reviewer 4 Report

Comments and Suggestions for Authors

This is a very comprehensive summary of the field of research dedicated to antibody-based treatment of Ebola virus disease. The authors have clearly conducted a very thorough review of the literature, and the supplementary tables at the end are a testament to this.

Overall this review is nicely structured, moving from polyclonal to monoclonal antibodies (MAbs) and then MAbs drugs. Some further and clearer subdivision of each paragraph would be appreciated though as currently several sections are very dense. Use of additional subtitles would help, instead of often beginning each paragraph with a brief mention of a specific study which the reader might not be familiar with (e.g. line 138). The pattern of starting paragraphs this way actually continues throughout, e.g. line 305. Please review.

To make tables S2 and S3 more useful for comparisons with future studies, please include a column that states % survival of control subjects.

Minor comments:

Figure 1 – briefly mention how an outbreak is defined.

Line 63 – ‘demonstrates the effectiveness of the method’ is immediately contradicted by the next statement. Recommend replacing ‘demonstrates’ with ‘suggests’, then continuing to explain caveats.

Line 71 – ‘which is 7% higher..’ is an unnecessary statement here. Are there any supporting statistics to indicate this is significant?

Line 79 – please clarify what the control moneys were. Group 1? Or 3rd group?

Line 88 – clarify what is the ‘drug’. Hyperimmune serum?

Line 154 – advise reordering to ‘..the only primate that survived..’

Line 174 – please omit ‘deteriorates the immune system’. This is not clear. Also the conclusion in lines 177-179 of this section is debatable. This doesn’t make sense in the context of live virus vaccination.

Figure 2 – by ‘vulnerability’, do you mean most targeted by GP-specific antibodies?

Line 395 – do you mean staggered instead of staged?

Line 574 – please provide numbers of animals when describing these animals, ‘one animal’ could be 100%, and ‘part of experimental animals’ needs grammatical revision. Including numbers and % of animals need to be applied to later experiments in this section too.

Line 610 – frequency of administration should also include time interval information.

Line 667 – this is a critical conclusion, are there any suggestions as to why no effect was seen? Was there any virus strain variation?

Line 745 – whilst this statement mentions tremendous success, the sentence immediately following states a series of issue. This is confusing.

Line 756 – please elaborate, why should studies proceed? Can you make suggestions as to what should be done in the future?

Final conclusion sentence – unusual to end on a new reference; suggest discussing this particular reference in the main text as it is agreed this is an intriguing phenomenon.

Table S3 – incorrect spelling of unknown in column 3.

Please check scientific numbers throughout e.g. Line 160 (and 435) – 10^3 PFU or 103 PFU?

Author Response

The team of authors would like to thank the reviewer for high evaluation of our work and making important comment.

  1. Overall this review is nicely structured, moving from polyclonal to monoclonal antibodies (MAbs) and then MAbs drugs. Some further and clearer subdivision of each paragraph would be appreciated though as currently several sections are very dense. Use of additional subtitles would help, instead of often beginning each paragraph with a brief mention of a specific study which the reader might not be familiar with (e.g. line 138). The pattern of starting paragraphs this way actually continues throughout, e.g. line 305. Please review.

Thank you for your comment!

Line 138. An experiment by Jahrling et al. was unsuccessful (Table S2) [23].

At the same time, a similar approach implemented by Jahrling et al. was unsuccessful

Line 305. Bornholdt et al. achieved impressive results [53].

Using the approaches previously developed for the search and generation of broadly neutralizing antibodies, including those against HIV-1, Bornholdt et al., obtained several hundred antibodies based on the set of antibodies from a convalescent donor. Such a large experimental scale made it possible to not only obtain novel anti-body variants with excellent characteristics but also elucidate the general pattern of epitope specificity against EBOV GP in an organism [59].

Line 345. D. Corti et al. generated highly effective monoclonal antibodies mAb100 and mAb114.

Based on the pool of antibodies from a Kikwit outbreak survivor (DRC, 1995), Corti et al. generated highly effective monoclonal antibodies mAb100 and mAb114 [65]

Line 494. Bornholdt et al. also managed to identify cross-neutralizing antibodies among the obtained variants

Screening of the variety of antibodies obtained from a survivor of the 2014 Ebola virus outbreak [59] against rVSV BDBV, SUDV, and EBOV made it possible to select cross-reactive variants, with some of them being broadly neutralizing.

  1. To make tables S2 and S3 more useful for comparisons with future studies, please include a column that states % survival of control subjects.

Thank you for your comment! We include a column that states % survival of control subjects into tables S2 and S3

  1. Figure 1 – briefly mention how an outbreak is defined.

Thank you for your comment! We have included the following information in the caption to Figure 1:

The chart is based on the data of the US Centers for Disease Control and Prevention [https://www.cdc.gov/vhf/ebola/history/chronology.html]. An outbreak was defined as any reported case of one of the Ebolaviruses.

  1. Line 63 – ‘demonstrates the effectiveness of the method’ is immediately contradicted by the next statement. Recommend replacing ‘demonstrates’ with ‘suggests’, then continuing to explain caveats.

Thank you for your comment! We replaced the word 'demonstrates' with 'suggests' in the specified line.

  1. Line 71 – ‘which is 7% higher..’ is an unnecessary statement here. Are there any supporting statistics to indicate this is significant?

Thank you for your comment! Indeed, the difference between the groups is not significant. We removed the phrase ‘which is 7% higher..’ from the sentence.

The survival rate in the control group (no serum transfusion) was 62% (Table S2).

  1. Line 79 – please clarify what the control moneys were. Group 1? Or 3rdgroup?

Thank you for your comment! We have clarified the control group number.

All control monkeys (group 3) died, while only one animal receiving antibody preparation together with infection died in the experimental group [10].

  1. Line 88 – clarify what is the ‘drug’. Hyperimmune serum?

Thank you for your comment! We have clarified the proposal.

The protective effect of the drug based on hyperimmune equine serum was observed in guinea pigs only [13].

  1. Line 154 – advise reordering to ‘..the only primate that survived..’

Thank you for your comment! We changed the word order in the sentence.

The only primate that survived belonged to the group receiving serum from a survivor of SUDV infection [25].

  1. Line 174 – please omit ‘deteriorates the immune system’. This is not clear. Also the conclusion in lines 177-179 of this section is debatable. This doesn’t make sense in the context of live virus vaccination.

Thank you for your comment! We have changed the wording of this sentence in the article.

Immunization with inactivated virus appears to induce antibody production with ad-verse effects such as antibody-dependent enhancement (ADE).

  1. Figure 2 – by ‘vulnerability’, do you mean most targeted by GP-specific antibodies?

Yes, “regions of vulnerability” is the most targeted regions by GP-specific antibodies

  1. Line 395 – do you mean staggered instead of staged?

Thank you for your comment! We have used staggered instead of staged.

  1. Line 574 – please provide numbers of animals when describing these animals, ‘one animal’ could be 100%, and ‘part of experimental animals’ needs grammatical revision. Including numbers and % of animals need to be applied to later experiments in this section too.

Thank you for your comment! We have added the number of animals.

The results of this analysis turned out to be rather ambiguous. In the first experiment, two (100%) NHPs treated with MB-003 survived, while one (100%) control NHPs died. In the second experiment, one (20%) animal receiving MB-003 died, and four (80%) animals receiving MB-003 survived. The one (100%) control animal survived as well. Finally, two (34%) experimental animals receiving MB-003 died while four (67) survived in the third experiment (all control animals are dead).

  1. Line 610 – frequency of administration should also include time interval information.

Thanks for your comment! Information about time interval has been added to the proposal.

The frequency of drug administration, which equals three for ZMAb (1,4,7 or 2,5,8 days after infection), is the same for ch133+ch226 but higher than that of KZ52 (two injections) [89].

  1. Line 667.– this is a critical conclusion, are there any suggestions as to why no effect was seen? Was there any virus strain variation?

Thanks for your comment! The authors of the paper [97] claimed statistical significance of the results. The reason why mortality among patients who received ZMapp was 22% in the trial [95] and 50% in the trial [97] is unclear. It is worth noting that in the study [95] the result did not meet the pre-specified statistical threshold for efficacy. The authors [97] suggest that the CFR may be affected by the drug administration schedule. For example, for Ebanga and Inmazeb, the drug is administered in full at the beginning of treatment. ZMapp and remdesivir were administered gradually over several days.

  1. Line 745 – whilst this statement mentions tremendous success, the sentence immediately following states a series of issue. This is confusing.

Thanks for your comment! We have rewritten and supplemented the conclusion section.

  1. Line 756 – please elaborate, why should studies proceed? Can you make suggestions as to what should be done in the future?

Thanks for your comment! We have rewritten and supplemented the conclusion section.

  1. Final conclusion sentence – unusual to end on a new reference; suggest discussing this particular reference in the main text as it is agreed this is an intriguing phenomenon.

Thanks for your comment! We have rewritten and supplemented the conclusion section.

  1. Table S3 – incorrect spelling of unknown in column 3.

Thanks for your comment! We have corrected the spelling of the word "unknown" in Table S3.

  1. Please check scientific numbers throughout e.g. Line 160 (and 435) – 10^3PFU or 103 PFU?

Thank you for your comment! We have corrected the incorrectly written numbers in the text.

Round 2

Reviewer 2 Report

Comments and Suggestions for Authors

No further comments